# Exploration of the Binding Mechanism of Cyclic Dinucleotide Analogs to Stimulating Factor Proteins and the Implications for Subsequent Analog Drug Design

**DOI:** 10.3390/biom14030350

**Published:** 2024-03-14

**Authors:** Shu-Wei Yuan, Hong-Ling Shi, Mu-Ran Fu, Xi-Chuan Zhang, Xiao-Qi Xi, Yao Wang, Tai-Song Shen, Jin-Liang Ma, Cun-Duo Tang

**Affiliations:** 1Henan Provincial Engineering Laboratory of Insect Bio-Reactor and College of Life Science and Agricultural Engineering, Nanyang Normal University, 1638 Wolong Road, Nanyang 473061, China; yuanshuwei@stu.htu.edu.cn (S.-W.Y.); 20142041@nynu.edu.cn (H.-L.S.); 18680140314@nynu.edu.cn (M.-R.F.); 2023086001018@nynu.edu.cn (X.-C.Z.); 2022086004013@nynu.edu.cn (X.-Q.X.); 2022086001025@nynu.edu.cn (Y.W.); 2022086001020@nynu.edu.cn (T.-S.S.); 2Henan Key Laboratory of Organic Functional Molecule and Drug Innovation, School of Chemistry and Chemical Engineering, Henan Normal University, 46 Jianshe East Road, Xinxiang 453007, China; 3School of Bioengineering, Dalian University of Technology, 2 Linggong Road, Dalian 116024, China

**Keywords:** cyclic dinucleotides, analogs, stimulator of interferon genes, adjuvants, second messenger, computer simulations

## Abstract

Cyclic dinucleotides (CDNs) are cyclic molecules consisting of two nucleoside monophosphates linked by two phosphodiester bonds, which act as a second messenger and bind to the interferon gene stimulating factor (STING) to activate the downstream signaling pathway and ultimately induce interferon secretion, initiating an anti-infective immune response. Cyclic dinucleotides and their analogs are lead compounds in the immunotherapy of infectious diseases and tumors, as well as immune adjuvants with promising applications. Many agonists of pathogen recognition receptors have been developed as effective adjuvants to optimize vaccine immunogenicity and efficacy. In this work, the binding mechanism of human-derived interferon gene-stimulating protein and its isoforms with cyclic dinucleotides and their analogs was theoretically investigated using computer simulations and combined with experimental results in the hope of providing guidance for the subsequent synthesis of cyclic dinucleotide analogs.

## 1. Introduction

Stimulator of interferon genes (STING) plays a crucial role in innate immunity and belongs to the class of articulatory proteins, especially in coordinating the body’s response to cytoplasmic DNA, be it pathogenic, auto-, or tumor DNA. It is widely expressed in immune cells, such as dendritic cells, macrophages, and B-cells, as well as in endothelial cells, among others. The activation of STING occurs through the interferon regulatory factor (IRF) pathway to produce type I interferons, such as IFN-α and IFN-β, and through the nuclear factor κ-activated B-cell light-chain enhancer (NF-κB) pathway to produce pro-inflammatory cytokines, such as tumor necrosis factor α (TNF-α) and interleukin 1β (IL-1β).

Up to now, there have been four natural cyclic dinucleotides found in humans, namely 3′,3′-c-di-AMP [1], of microbial origin, 3′,3′-cGAMP [2], 3′,3′-c-di-GMP [3], and 2′,3′-cGAMP, which were found in humans. 2′,3′-cGAMP synthesized in humans by cyclic guanosine monophosphate aden-sine monophosphate synthase [4,5,6]. The cascade reaction of 2,3 cyclic guanosine monophosphate aden-sine monophosphate synthase (GAS) synthesized by cyclic guanosine monophosphate aden-sine monophosphate synthase (GAS) in the human body ultimately induces the production of type I interferon and other cytokines to initiate the immune response, and the cyclic di-nucleotide binds to the stimulator of interferon genes (STING) to activate the downstream signaling cascade reaction to eventually induce the production of type I interferon and other cytokines to initiate the immune response, and the cGAMP is found in the human body, and it is also known as 2′,3′-cGAMP. The cGAS-STING signaling pathway is a core component of the human nucleic acid immune defense mechanism. cGAS is activated and catalyzes the generation of cyclic GMP-AMP, 3′,3-cdi-AMP, 3′,3′-d-GMP, or 3′,3′-cGAMP produced by infectious microorganisms. cGAS is also activated by the combination of dsDNA released by pathogenic microorganisms in the body and cyclic GMP-AMP dinucleotide GMP-AMP (2′,3′-cGAMP), which directly binds to the endoplasmic reticulum (ER) receptor protein STING (IFN gene stimulator) and locks it into the V-shaped ligand-binding pocket of the STING dimer, causing a conformational change and oligomerization of STING. STING oligomers then leave the endoplasmic reticulum and travel to the endoplasmic reticulum-Golgi intermediate compartment (ERGIC) by an unknown mechanism. The transfer of STING from the endoplasmic reticulum to the Golgi is necessary for the palmitoylation of two cysteine residues of STING in the Golgi. Palmitoylation further enhances STING oligomerization and activates TANK-binding kinase 1 (TBK1) and activates signaling pathways. Indeed, STING mediates the transcription of type I interferon (IFN) and other inflammatory cytokines by recruiting downstream TBK1 kinases to phosphorylate the transcription factor IRF3. Activated IRF3 enters the nucleus of the cell and mediates the transcription of type I interferon (IFN) and other inflammatory cytokines in response to microbial invasion [7,8].

Currently, a total of four major non-synonymous variants of hSTING have been identified in the scientific community, which are haplotypes G230A and R293Q, containing a single amino acid substitution, and R232H and R293Q, consisting of two co-segregating substitutions (named AQ), and three substitutions—R71H, G230A, and R293Q (named HAQ) (Figure 1A). It is shown that R71H, G230A, and R293Q (HAQ) occur most frequently in humans by analyzing the 1000 Genomes Project’s single nucleotide polymorphism (SNP) data, which showed that R71H-G230A-R293Q (HAQ) had the highest incidence, occurring in about one-fifth of humans; R232H occurred in 13.7%; G230A-R293Q (AQ) occurred in 5.2%; and R293Q had the lowest incidence, occurring in only 1.5% of humans [9,10].

The first cyclic dinucleotide was discovered back in the 1980s, followed by the discovery of three other cyclic dinucleotides in microbial species, but this did not attract much attention from the scientific community. It was not until the early part of this century that a group of researchers from the University of Texas Southwestern Medical Center demonstrated that cyclic dinucleotides were immunotransmitters of the cGAS-STING signaling pathway. A large number of researchers working on cyclic dinucleotide analogs or non-nucleoside cGAS-STING agonists have emerged, but so far, only a few of them have been able to enter clinical trials [11,12] (Table 1).

In fact, the current structural modifications of cyclic dinucleotides are mainly focused on the following three parts, which are the thiolation or substitution of the phosphodiester bond part, the removal or substitution of the hydroxyl group in the ribonucleic ring part, and the modification of the base part (Figure 2) [13,14,15].

In this study, we selected representative compounds of each modification for molecular docking in the hope of providing insights into the mechanism by which the human stimulator of interferon genes protein (hSTING) can bind cyclic dinucleotides (CDNs) to activate the mechanism of production of type I interferons and inflammatory cytokines to explain the computerized means, such as molecular docking, and provide insights into the subsequent synthesis of agonists.

## 2. Materials and Methods

### 2.1. Web Servers and Software

Basic Local Alignment Search Tool (BLAST, https://blast.ncbi.nlm.nih.gov/Blast.cgi, (accessed on 3 May 2023)) (for USA) was analyzed for searching protein crystal structures. PubChem (https://pubchem.ncbi.nlm.nih.gov/, (accessed on 3 May 2023)) (for USA) was used for small-molecule stereo structure findings. EasyModeller2.0 (for USA) [16], Swiss-Model (for Swiss Confederation) [17], AlphaFold2 (for USA) [18], and RCSB PDB (https://www.rcsb.org/, (accessed on 3 May 2023)) (for USA) were used to perform multi-template analysis of the hSTING protein’s 3D structure for multi-template homology modeling. AutoDock Vina (for USA) [19] was used for STING protein docking with small molecules. Molecular dynamics simulations of the proteins were performed using the GROMACS 4.5.4 program [20], and PyMOL was used to observe, analyze, and map the 3D structures of the hSTING proteins and their small molecules. Discover Studio was used to observe the number and type of spatial forces binding the small molecules to the hSTING proteins.

### 2.2. Homology Modeling of These hSTING

In this study, in order to obtain more accurate 3D structures, we used AlphaFold2 and RCSB PDB (https://www.rcsb.org/) to simulate a number of 3D structures with a high degree of similarity to the parental species, optimized these models and their amino acid sequences using EasyModeller2.0, and finally obtained more accurate and reliable 3D structures for subsequent use. We then optimized these models and their amino acid sequences using EasyModeller2.0 and finally obtained more accurate and reliable 3D structures for subsequent use. The modeling parameters are shown in Appendix A.

### 2.3. Molecular Docking

According to our previous study, the three amino acid residues in hSTING that are associated with small-molecule drug binding are Tyr167, Arg232, and Arg238. In this study, we used the web site PubChem (https://pubchem.ncbi.nlm.nih.gov/) for the small-molecule stereo structure finding [21] and the Chem3D Ultra 8.0 for energy minimization of the small molecules [22]. Molecular docking simulations of flexible ligands were performed by genetic algorithms using the program AutoDock Vina to locate suitable binding sites. The scoring functions for docking calculations are shown in Appendix A.

### 2.4. Molecular Dynamics Simulations

We attempted to perform 100 ns MD simulations of the parental and four isoforms by GROMACS 4.5.4 at 300 K ambient temperature, OPLS-AA/L all-atom force field, NaCl equilibrium protein charge, NVT equilibrium, and NPT equilibrium. The stable RMSD value of the last 10 ns ensured the reliability of the MD trajectories. RMSD, as an indicator for evaluating thermal fluctuations, is negatively correlated with the thermal stability of proteins [23,24].

## 3. Results and Discussion

### 3.1. Sequence Analysis of hSTING Protein Parents and Their Isoforms

According to the SNP analysis, hSTING has four major isoforms: R232H, R293Q, G230A-R293Q (AQ), and R71H-G230A-R293Q (HAQ). We subjected them to multiple-sequence comparative analysis, and the results, as shown in Figure 1B, showed that the five amino acid sequences are highly homologous, which indicates that the four loci of hSTING are highly conserved. In addition, we input the amino acid sequences of hSTING parents and the four isoforms into AlphaFold2 [25] and SWISS-MODEL (https://swissmodel.expasy.org/, (accessed on 3 May 2023)) for homology modeling, and the similarity between the obtained models and their sequence features was higher than 98%. At the same time, we used RCSB PDB (https://www.rcsb.org/) for homology modeling, and the five models obtained, including 6NT5, 8GT6, 8GSZ, 8IK3, and 7SII, were all more than 99% similar to their sequences. Based on the above conditions, we finally imported the amino acid sequences into EasyModeller2.0 and imported the amino acid sequences into SWISS-MODEL (https://www.rcsb.org/), and the models were all ranked according to the similarity of the amino acid sequences to their sequences, in ascending order. We imported EasyModeller2.0, optimized them again, and finally obtained a more accurate and reliable three-dimensional structure for molecular docking.

### 3.2. Molecular Dynamics Simulation Results

To explore the molecular basis of the different temperature properties of these hSTINGs, we performed 100 ns molecular dynamics simulations at room temperature (300 K). The RMSD values (Figure 3A), RMSF values (Figure 3C), and radius of gyration values (Figure 3D) of hSTING-WT, hSTING-R232H, hSTING-R293Q, hSTING-AQ, and hSTING-HAQ were calculated using the g-rms command. The distribution of RMSD values of hSTING-WT, hSTING-R232H, hSTING-R293Q, hSTING-AQ, and hSTING-HAQ was then statistically analyzed using Origin 9.0 (Figure 3B).

As shown in Figure 3A, the RMSD values of hSTING-HAQ were smaller than those of hSTING-R232H, hSTING-R293Q, hSTING-AQ, and hSTING-WT for a considerable period of time. hSTING-HAQ’s RMSD values were mostly centered around 1.3 nm, which was significantly lower than those of other hSTING isoforms. This result fully demonstrated that hSTING-HAQ was more stable at room temperature compared with other isoforms. It has been clearly reported that the RMSD value is negatively correlated with the thermal stability of the protein. Therefore, these results initially revealed the molecular basis of the thermal stability of hSTING-HAQ compared with other isoforms [26].

As shown in Figure 3C, the RMSF values of hSTING-HAQ were much smaller than those of hSTING-R232H, hSTING-R293Q, hSTING-AQ, and hSTING-WT most of the time, except for the ministries at the loci shown in the figure. This result indicates that the degree of freedom of individual amino acid residues within hSTING-HAQ is lower than that of the other isoforms.

The global radius of gyration is a parameter that caters for the conformational size of the entire molecule, which is calculated from the average distance of all atoms around the center of mass of the molecule. The global radius of gyration can help to analyze the overall conformational features and size of the molecule, with a smaller radius indicating a stronger densification, i.e., the more stable the protein structure will be. In Figure 3D, we can see that the values of hSTING-HAQ and hSTING-R293Q are much smaller than the other three hSTING proteins in the late stage of the radius of gyration, and the value of hSTING-HAQ is slightly smaller than that of hSTING-R293Q after 70 ns, which proves that the protein structure of hSTING-HAQ can still be more stable under prolonged simulation.

### 3.3. Molecular Docking

#### 3.3.1. Natural Cyclic Dinucleotides

In 2013, Gao et al. [4] identified cyclic GMP-AMP (cGAMP) as the postnatal animal second messenger that triggers the interferon response. In their study, c[G(2′,5′) pA(3′,5′)p] has been identified as a founding member of the postnatal animal 2′,5′ cyclic heterodinucleotide second messenger family. We subjected them to molecular docking, and the results are shown in Figure 4A. The cyclic dinucleotide small molecules not only formed more intermolecular forces with the three amino acid residues mentioned above, but also with several amino acid residues, such as Ile170, Ala233, Tyr240, etc. And the hydrogen bonding lengths were generally shorter, with a total binding affinity of −7.5 kcal/mol. This result can indicate that the compound can stimulate the activation of cGAS-STING signaling pathways and produce type I interferon as the second messenger.

The corresponding hSTING-R232H isoform mutated its arginine at position 232 to histidine relative to the parent, and we attempted to molecularly dock it. The results are shown in Figure 4B: the number of intermolecular forces formed between histidine at position 232 and the small molecule is much greater than that of the parent, and the total binding affinity is −7.8 kcal/mol, which is slightly smaller than that of the parent. The results suggest that this small molecule may be more suitable for stimulating the hSTING-R232H isoform signaling pathway relative to the parent.

#### 3.3.2. Base Position Modified Cyclic Dinucleotide Analogs

In 2007, Wang et al. [27] found that it was possible to obtain seven more CDNs containing the canonical 3′-5′ phosphodiester bond linkage by combining four naturally occurring ribonucleotides in addition to the three naturally occurring CDNs.

The authors synthesized all ten possible combinations of 3′-5′-linked CDNs. A novel CDN c-AMP-CMP was identified as a new immunostimulant.

However, we did not find the corresponding binding site in the original grid box size, and then we tried to double the size of the grid box and found the corresponding binding site. And the total binding affinity was only −7.6 kcal/mol. From the molecular docking results (Figure 5), we can see that, compared with the natural cyclic di-nucleotide, it is inferior to the natural cyclic di-nucleotide in terms of the number of intermolecular forces formed and the bond length. No intermolecular forces are even formed with Arg232. Therefore, we speculated that the induction of type I IFN responses by the CDNs is much lower compared with natural cyclic dinucleotides. The reliability of our guess and the docking results is confirmed by the final article conclusion, in which the authors describe the finding of c-AMP-CMP as a weak immunostimulatory factor that produces only a weak type I IFN response in mouse macrophages through the same signaling axis, as already described for CDNs.

#### 3.3.3. Ribose Position-Modified Cyclic Dinucleotide Analogs

The replacement of one or two hydroxyl groups at the 2′ or 3′ positions of nucleosides (nucleotide) with halogenated elements such as fluorine and chlorine is a common strategy in related drug research. Zhou et al. successfully synthesized 2′-F-c-di-GMP using N-isobutyryl-2′-F guanosine phosphoramidite as a starting material by the Jones one-pot method in 2013 [28]. We performed molecular docking for structural observation (Figure 6), and the total binding affinity between this small molecule and the receptor protein was about -6.6 kcal/mol, approximating the natural cyclic dinucleotide. We can see that more than 20 hydrogen bonds and other intermolecular forces were formed between the compound and many amino acid residues, such as Arg232 and Arg238. And more than ten hydrogen bonds were formed between the compound and Arg232 residues alone. While such a large number of intermolecular forces prompted the small molecule to bind to the receptor tightly, competitive inhibition may occur between the sites, which may reduce the tightness of the compound to the receptor protein and thus affect the biological activity of the small molecule. From the perspective of disrupting bacterial biofilm formation, small molecules can be used to inhibit c-di-GMP synthesis or compete with c-di-GMP-binding effector molecules. The discovery that 2′-F-c-di-GMP is a potent inhibitor of c-di-GMP synthesis laid the groundwork for the preparation of cellularly permeable analogs of 2′-F-c-di-GMP that can be used to interfere with c-di-GMP-signaling permeable analogs lays the foundation.

#### 3.3.4. Phosphodiester Bond Position-Modified Cyclic Dinucleotide Analogs

The sulfur modification of the phosphodiester bond portion was developed and perfected during the research of oligodeoxynucleotide drugs, and several sparsely substituted oligodeoxynucleotide drugs have been approved for clinical use because sulfurization improves the resistance of oligodeoxynucleotides to degradation by phosphodiesterase and nuclease [29]. Sulfurization modification of the phosphodiester bond of CDNs is an important way of improving their stability, lipophilicity, and the modulation of their bioactivities [30].

Due to the chiral nature of the phosphorothioate diester bond, thio–CDN is a mixture of various diastereoisomers. GAFFNEY et al. [31] were the first to report the synthesis of thio–CDN by the H-phosphite/phosphoramidite method and used high-performance liquid chromatography (HPLC) to isolate and obtain a single chiral isomer of pure CDN analogs. In order to investigate the effect of chiral pure thiocyclic dinucleotide analogs on their biological activities, CORRALES et al. synthesized and isolated chiral pure CDN analogs using the above method, and we molecularly docked them with the hSTING parent (Figure 7) and isoforms, respectively. Although the binding affinity was equal to −6.9 kcal/mol, which was slightly higher than natural cyclic dinucleotides, the conformation was similar to the natural cyclic dinucleotide. However, this configuration has a relatively close spatial distance to the Arg238 residue, which can form various intermolecular forces except for hydrogen bonding. But surprisingly, according to the molecular docking results, this compound may have a relatively good stimulatory effect on all five existing hSTINGs. And the results of subsequent cellular experiments conducted by the authors of the paper showed that the activity of thio-Rp and Rp-2′,3-c-di-AMI was significantly improved and active against five known human STING variants. The problem of limiting CDNs from being suitable for clinical development due to the presence of a significant frequency of hSTING alleles refractory to these structures in the population has been addressed to some extent.

#### 3.3.5. Multisite-Modified Cyclic Dinucleotide Analogs

Because of the complexity of the mechanism of action of CDN, there is no certainty about the conformational relationship between CDN and its analogs. Based on the satisfactory biological activities of CDN as well as its analogs in animal tumor immunotherapy and as immune adjuvants, many research institutes and pharmaceutical companies around the world have invested heavily in the multisite modification of CDN in order to obtain compounds with better activities suitable for clinical development.

In 2021, Li et al. [32] designed and synthesized a novel STING agonist, CDG^SF^, which was modified by phosphorothioate and fluorine. The author team suggests that the phosphorothioate in CDG^SF^ may be a covalent binding site. In particular, immunization against SARS-CoV-2 spiking proteins using CDG^SF^ as an adjuvant elicited abnormally high antibody titers and robust T-cell responses, overcoming the drawbacks of aluminum hydroxide. The authors demonstrated for the first time that STING agonists can serve as excellent adjuvants for SARS-CoV-2 vaccines, overcoming the drawbacks of aluminum hydroxide for SARS-CoV-2 vaccines. These results highlight the therapeutic potential of CDG^SF^ in cancer immunotherapy and the adjuvant potential of STING agonists in SARS-CoV-2 vaccine preparation.

We attempted to molecularly dock it, and the total binding affinity was −6.9 kcal/mol, which approximated the natural cyclic dinucleotide. As shown in Figure 8A, the amino acid residues are tightly bound to the phosphorothioate site of this cyclic dinucleotide analog, forming multiple hydrogen bonds, a result that confirms the authors′ speculation in the paper that the sulfur in the phosphorothioate CDN may be a covalent coupling site. The authors again used a melanoma mouse model for validation in the article, and the experimental results demonstrated that CDG^SF^ treatment significantly inhibited melanoma growth.

Merck & Co, Inc. (Rahway, NJ, USA) designed and synthesized the stereochemically and structurally complex cyclic dinucleotide-based interferon gene stimulator (STING) agonist, MK-1454, in March 2022, and the evaluation of its biochemical affinity and cellular potency, as well as computational, structural, and biophysical characterization, were applied to influence the design and optimization of the novel STING agonist, which led to the discovery of MK-1454 as a molecule with appropriate properties for clinical development [33,34].

We attempted to molecularly dock it with a total binding affinity of only −7.2 kcal/mol, approximating a natural cyclic dinucleotide. As shown in Figure 8B, the amino acid residue Arg238 forms hydrogen bonds with small molecules, while also forming unfavorable positive–positive bonds with carbon–carbon double bonds. But the unique aromatic ring-containing structure of Tyr167 provides favorable conditions for π-π stacking with small molecules while Thr263, Ser162, and other amino acid residues with the small molecules form a variety of intermolecular forces, like carbon-hydrogen bonding. Under the combined effect of these intermolecular forces, the small molecule is tightly bound to hSTING. In the authors’ paper, when administered intratumorally to immunocompetent mice with homozygous tumors, MK-1454 exhibited potent tumor cytokine upregulation and potent antitumor activity. Therefore, we hypothesize that MK-1454 has a broader application prospect and deserves to be further explored by the relevant researchers.

Wei Xie et al. [35], 2023, recently reported a new class of chemically synthesized sugar-modified 2′,3′-cGAMP analogs containing arabinose and xylose derivatives capable of binding mouse and human STING alleles with high affinity. We attempted to perform molecular docking on it, with a total binding affinity of only −7.3kcal/mol, similar to natural cyclic dinucleotides, as shown in Figure 8C. The intermolecular forces formed by this compound with surrounding amino acid residues are similar to those of natural cyclic dinucleotides, but the 2′,3′-cGAMP analogues modified with arabinose and xylose have opened up a new strategy for overcoming the inherent nuclease-mediated fragility of natural ribonucleotides and have the additional advantage of high conversion potential as a cancer treatment and vaccine adjuvant. The author also tested the activity of this small molecule in the article, and the analogue showed significant resistance to hydrolysis-mediated ENPP1 and increased stability in human serum while retaining the induction of IFN- β secretion by human THP-1 cells, similar to 2′,3′-cGAMP. The modification of 2′,3′-cGAMP analogues with arabinose and xylose has opened up a new strategy.

#### 3.3.6. Cyclic Dipeptide Nucleic Acids and Their Analogs

Based on the successful precedent of peptide nucleic acids, the literature reports a cyclic dinucleotide analog with (2-aminoethyl)-glycine as the backbone [36]. But the authors did not analyze its bioactivity in the paper. And we subjected it to molecular docking (Figure 9), and its binding affinity was only −8.1 kcal/mol, which is less than that of the natural cyclic dinucleotide. But through the docking results, we found that its binding affinity with the receptor proteins Tyr167, Ser241, Arg232, and Thr263 could form one or two hydrogen bonds directly, and there was no other form of intermolecular force. The single mode of intermolecular force formation and the short bond lengths formed with amino acid active sites were favorable for enhancing the tightness of binding between the receptor proteins and small molecules. Therefore, we dare to speculate that this small molecule may be a more promising small-molecule STING protein agonist, which deserves continuous attention from subsequent researchers.

#### 3.3.7. Non-Nucleotide Small-Molecule STING Protein Agonists

DMXAA (5,6-dimethylstomathenone-4-acetic acid or vadimizan) [37], a stomathenone analog, was offered as a promising alternative to cGAMP but failed in phase III clinical trials with significantly fewer intermolecular forces and bond lengths than the classical agonist. This failure was attributed to the selective binding of DMXAA to mouse STING (mSTING) but not to human STING (hSTING) [38], despite the high sequence and structural similarity between the two protein molecules. (Figure 10A). The lack of recognition of hSTING by DMXAA has prompted the design of other derivatives; for example, α-Mangostin [39,40], a xanthone derivate, was found to bind to hSTING more strongly than mSTING (Figure 10B). The total binding affinity was −7.6 kcal/mol, which was even slightly better than that of natural cyclic dinucleotides. From the molecular docking results, multiple intermolecular forces, such as the formation of π-π stacking between Try167 and the aromatic ring of this derivative, may play a non-covalent bonding interaction as important as hydrogen bonding. In addition, there is also an interaction force between the alkyl group of this derivative and a number of amino acid residues, such as Pro264, Val239, Thr263, and so on. It has strong application prospects and utilization value [41,42].

In recent years, most research organizations have focused their attention on STING protein agonists with cyclic dinucleotide analogs. But Ramanjulu et al. [43] took a different approach, and in 2018, they discovered the first novel small-molecule non-nucleotide STING agonist to synergize the symmetry-related aminobenzimidazole (ABZI)-based symmetry-associated amino-benzimidazole (ABZI) compounds to create a link with STING and enhanced cellular function binding ABZI (di-ABZI). Intravenous injection of di-ABZI STING agonists into immunoreactive mice with established syngeneic colon tumors elicits potent antitumor activity with complete and durable tumor regression. Their findings represent a milestone in the rapidly evolving field of immunomodulatory cancer therapies. The results of its docking with the human-derived interferon gene-stimulating protein are shown in Figure 10C, with a total binding affinity of −6.3 kcal/mol, which approximates the natural cyclic dinucleotide. The formation of multiple intermolecular forces between the amino acid residues surrounding the protein’s active site allows it to bind tightly to it. But in the paper, the authors only demonstrated the antitumor activity of the agonist in mice. Based on the docking results, we speculate that the agonist’s anti-tumor activity in humans is not as good as that of natural cyclic dinucleotides. As the first non-nucleotide STING agonist discovered in humans, it is of great reference significance for the subsequent synthesis and screening of similar drugs.

Recently, Chen et al. [44] reported a novel monoamino benzimidazole (ABZI)-based analog D61 with nanomolar-level STING agonistic activity obtained through structural modification and optimization. We can find that by molecular docking (Figure 10D), its total binding affinity is −5.7 kcal/mol, which is at a high level, but the intermolecular forces formed between its key sites and the surrounding important amino acid residues are mainly hydrogen bonds. Multiple hydrophobic forces are formed between multiple sites and amino acid residues, such as alkyl and Π-alkyl bonds, which suggests that this small molecule is able to form good binding affinities with multiple important amino acid residues, and it is precisely because the natural properties such as the ease of hydrolysis of cyclic dinucleotides have hindered its development. The formation of hydrophobic forces can effectively solve this problem, which is of great significance for intermolecular binding. The shorter bond length indicates that the small molecule is closer to the amino acid residues and binds them more tightly. The experiments in the paper also confirmed our conjecture that D61 effectively inhibited tumor growth with good tolerance. This study of orally bioavailable aminobenzimidazole analogs expands the diversity of chemical structures of STING-mediated immunotherapeutic agonists and is an important drug candidate that deserves continued attention.

### 3.4. Discussion

At this stage, although CDNs and their analogs have shown potential as novel vaccine adjuvants and immunotherapies as important signaling molecules in the innate immune response, many therapeutic limitations have arisen for their real-life application due to the presence of multiple unfavorable external conditions, such as the intrinsic negative charge and the extracellular enzymes that cleave them. However, using the above studies, it is reasonable to assume that a deeper understanding of the nature of hSTING activation will help to establish the rationale for signaling with 2′, 3′-cGAMP and thus contribute to the design of relevant drugs targeting hSTING.

We summarized the results of numerous molecular docking, binding energy simulations, and literature analyses, and we found that Arg at position 232 among the active sites of human interferon gene stimulatory proteins plays a crucial role. All bacterial cyclic dinucleotides, c-di-GMP, c-di-AMP, and 3′, 3′-cGAMP, can be identified by Arg232 and activate IFN- β and NF-κb promoters. Whereas for the R232H variant, c-di-GMP partially reduces the c-di-GMP-induced response. The c-di-AMP and 3′, 3′-cGAMP pairs were significantly defective for the response. In the case of Arg238, upon agonist stimulation of the STING protein to produce a conformational change, the STING protein gradually closes and produces a “lid” to seal the binding pocket. Among the lid residues, cGAMP prefers to bind Arg238. During the simulation, the two Arg238 residues maintain strong cationic, hydrogen bonding, and attractive charge interactions with the cGAMP ligand. In addition, in molecular docking, we can also find the aromatic ring of Tyr167, which mediates key stacking interactions with the guanine or adenine rings of cGAMP and CDG in all small-molecule complex structures.

Based on the above molecular docking results, we believe that the three amino acids Arg232, Arg238, and Tyr167 in the STING protein play a decisive role in the screening of small molecules. And the three together form the most important component in the active center of the STING protein. But at the same time, we cannot ignore the amino acid residues such as Thr263, Gly166, Val239, and other nearby amino acid residues and small molecules directly formed by the force. It is the existence of these intermolecular forces that makes the molecule and the active site of the combination of the molecules more tightly bound. To a certain extent, we can even think that the more types of intermolecular forces formed by the small molecules and the surrounding amino acid residues, the shorter the force, like the length of the hydrogen bond, for the small molecule to become an effective agonist of the STING protein, the greater the potential for the application of a broader prospect.

For the small-molecule agonists covered in this paper, we calculated the average binding energies and standard deviations by analyzing their binding energies to different types of hSTING proteins using the most typical agonist, 2′, 3′-cGAMP, as a benchmark (Appendix A). From this analysis, we found that cyclic PNA agonists had the best binding effect on all five hSTING proteins, which was 6.4–12.2% higher than 2′, 3′-cGAMP, respectively. The mean value was 8.7% higher than 2′, 3′-cGAMP. And from the results of the analysis of the difference between the binding energies of various small-molecule drugs and 2′,3′-cGAMP (Figure 11), we can see that the ability of cyclic PNAs to bind hSTING proteins was significantly stronger than that of other small-molecule drugs. The molecular docking results above are consistent with the data analysis results, and based on the above analysis, we believe that cyclic PNAs and their analogues will become the hotspot of small-molecule agonist research in the future.

For small-molecule compounds, sp^2^ hybridization should be avoided as much as possible while possessing one or more torsion centers, and planar structures should also be avoided as much as possible so that small-molecule compounds can have more sufficient torsion angles and torsion space to bind to amino acid residues in the active site. This conclusion is limited to the backbone structure of small molecules. In addition, the introduction of aromatic regions into the side functional groups in addition to the backbone structure should not be overlooked in the pre-design process of small-molecule compounds for the formation of intermolecular forces in the form of π-π stacking with Tyr167, which is particularly important for the development of small-molecule agonists with acyclic dinucleotide analogs. Arg232 and Arg238 are basic amino acids, and the amino groups in the side-chain positions can easily form intermolecular interactions with the polar moieties (including hydroxyl groups). including hydroxyl groups) to form intermolecular forces, mainly in the form of hydrogen bonds. Therefore, we hypothesized that when designing small-molecule compounds, the number of hydroxyl groups in the small-molecule compounds should be increased appropriately, which is conducive to improving the activity of the compounds.

## 4. Conclusions

In summary, we summarized the five STING protein types discovered at this stage and analyzed their molecular dynamics at room temperature of 300 K. Based on the results of molecular dynamics simulation, we can see that the parents exhibit better physicochemical properties than the other four isoforms in most cases. Meanwhile, we also performed molecular docking on the small-molecule agonists of STING proteins that have been reported globally during the past few years. Based on the results of molecular docking, we tried to explore and describe the binding mechanism of small-molecule agonists to STING protein and to find out the preference of binding between the active site and the small molecule, which would be a revelation to the synthesis and design of the subsequent related drugs.

## Figures and Tables

**Figure 1 biomolecules-14-00350-f001:**
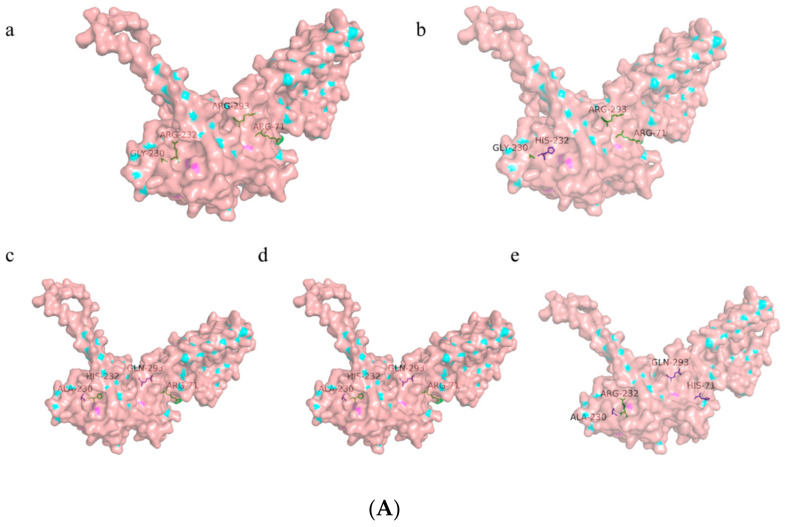
Homology analysis of hSTINGs. (**A**) hSTING-WT and four isoform protein surface models. (**a**–**e**) In order of hSTING-WT, hSTING-R232H, hSTING-R293Q, hSTING-AQ, and hSTING-HAQ. (**B**) Multiple-sequence comparison of hSTING-WT and four isoforms in this study.

**Figure 2 biomolecules-14-00350-f002:**
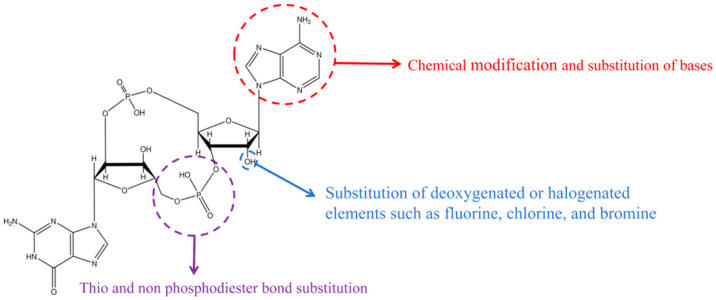
Structural modification sites of cyclic dinucleotides.

**Figure 3 biomolecules-14-00350-f003:**
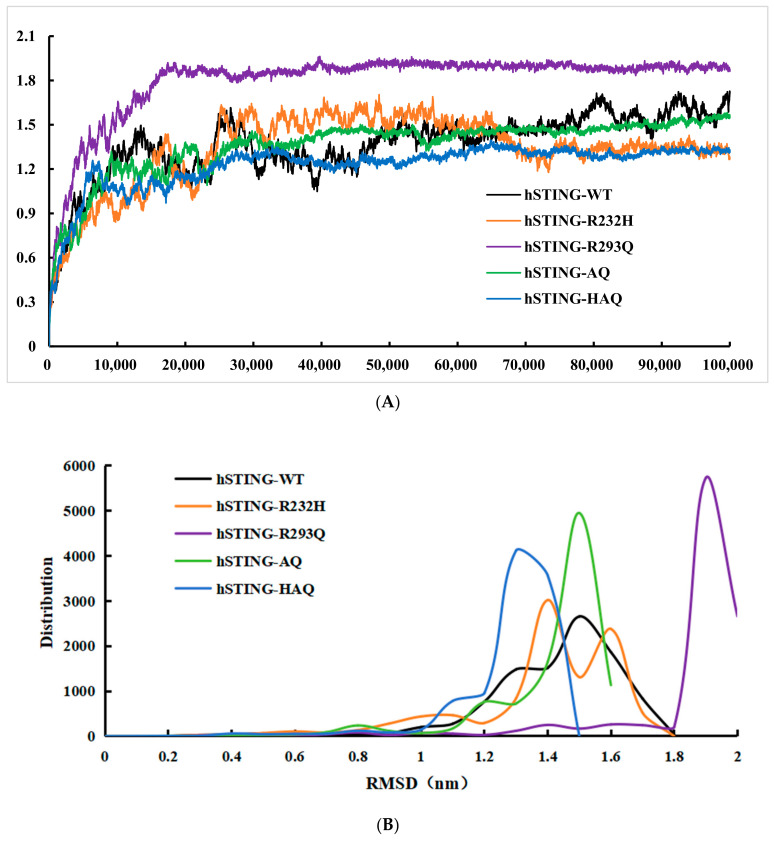
Molecular dynamics simulation results. (**A**) Calculation of RMSD values for hSTING-WT and the four subtypes. (**B**) Distribution of RMSD values for hSTING-WT and the four subtypes. (**C**) Calculation of RMSF values for hSTING-WT and the four subtypes. (**D**) hSTING-WT and four subtypes of global radius of gyration.

**Figure 4 biomolecules-14-00350-f004:**
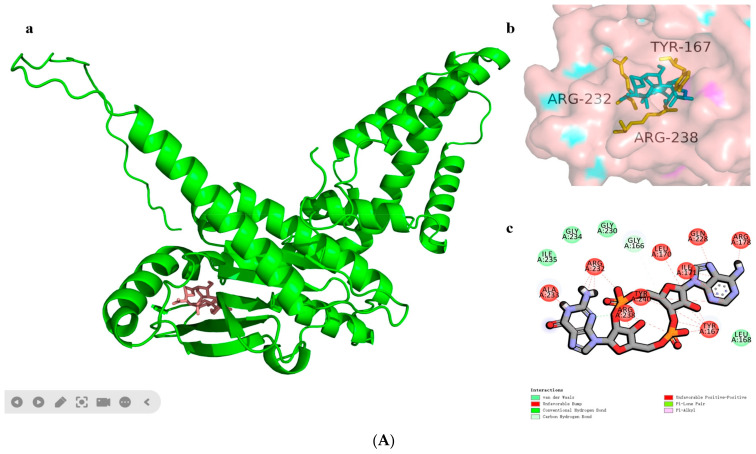
Docking of cGAMP to the hSTINGs. (**A**) Docking of cGAMP to the hSTING-WT reactive structural domain. (**B**) Docking of cGAMP to the hSTING-R232H reactive structural domain. The size of the grid box was set to 10 × 10 × 10. Molecular docking simulations were performed by genetic algorithms with flexible ligands in order to locate the appropriate binding orientation using the AutoDock 4.2 program. (**a**), docking macroscopic diagram, (**b**), docking microscopic diagram, (**c**), showing the types of spatial forces between amino acid residues and small molecular functional groups.

**Figure 5 biomolecules-14-00350-f005:**
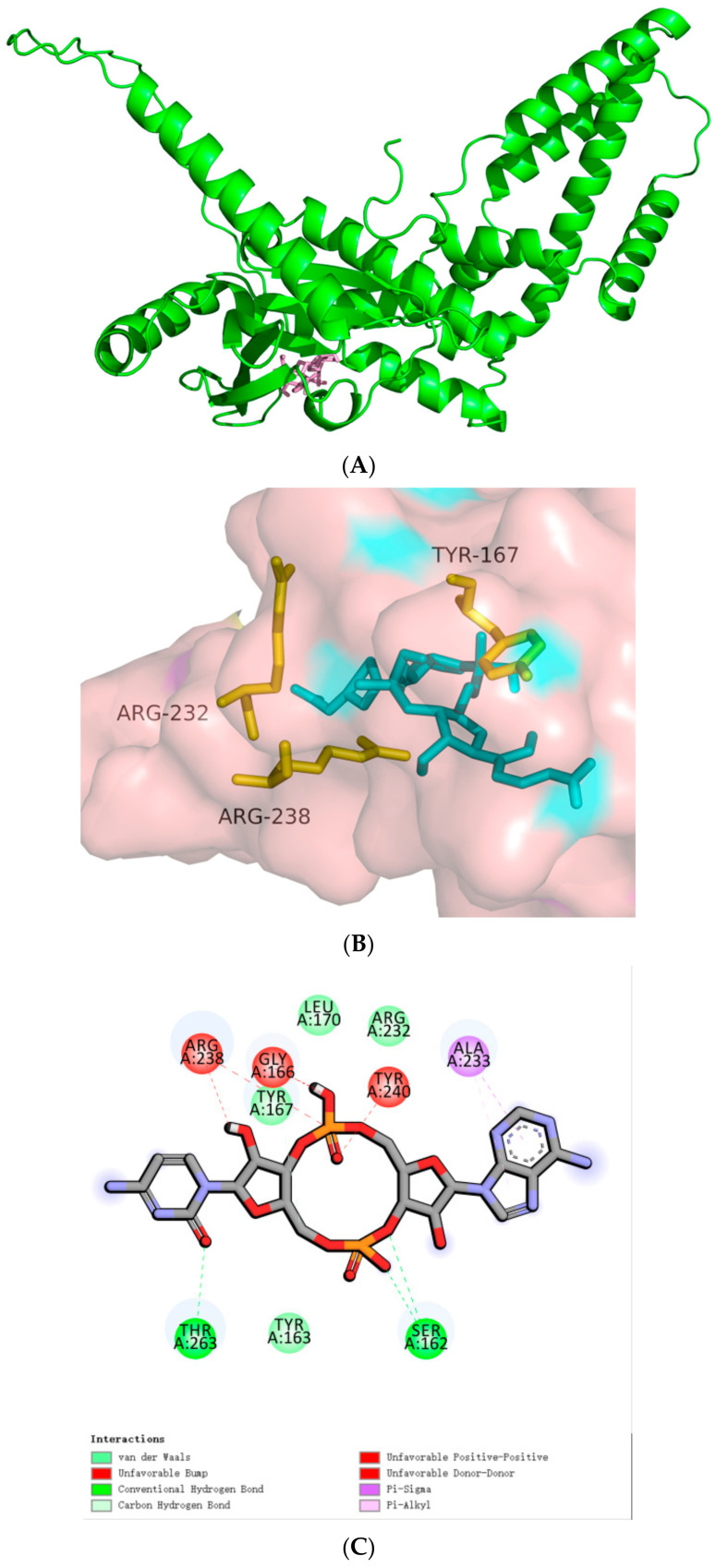
Docking of c-AMP-CMP to the hSTING-WT reactive structural domain. The size of the grid box was set to 15 × 15 × 15. Molecular docking simulations were performed by genetic algorithms with flexible ligands in order to locate the appropriate binding orientation using the AutoDock 4.2 program. (**A**) docking macroscopic diagram, (**B**) docking microscopic diagram, (**C**) showing the types of spatial forces between amino acid residues and small molecular functional groups.

**Figure 6 biomolecules-14-00350-f006:**
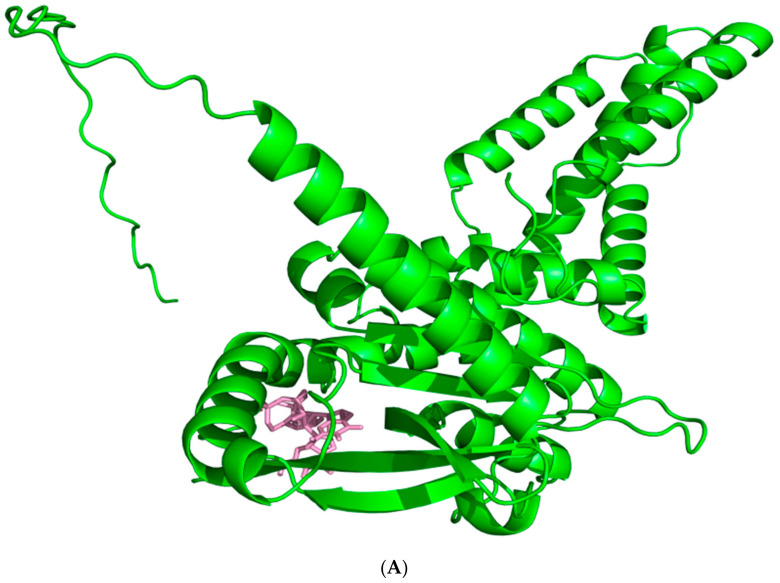
Docking of 2′-F-c-di-GMP to the hSTING-WT reactive structural domain. The size of the grid box was set to 10 × 10 × 10. Molecular docking simulations were performed by genetic algorithms with flexible ligands in order to locate the appropriate binding orientation using the AutoDock 4.2 program. (**A**) docking macroscopic diagram, (**B**) docking microscopic diagram, (**C**) showing the types of spatial forces between amino acid residues and small molecular functional groups.

**Figure 7 biomolecules-14-00350-f007:**
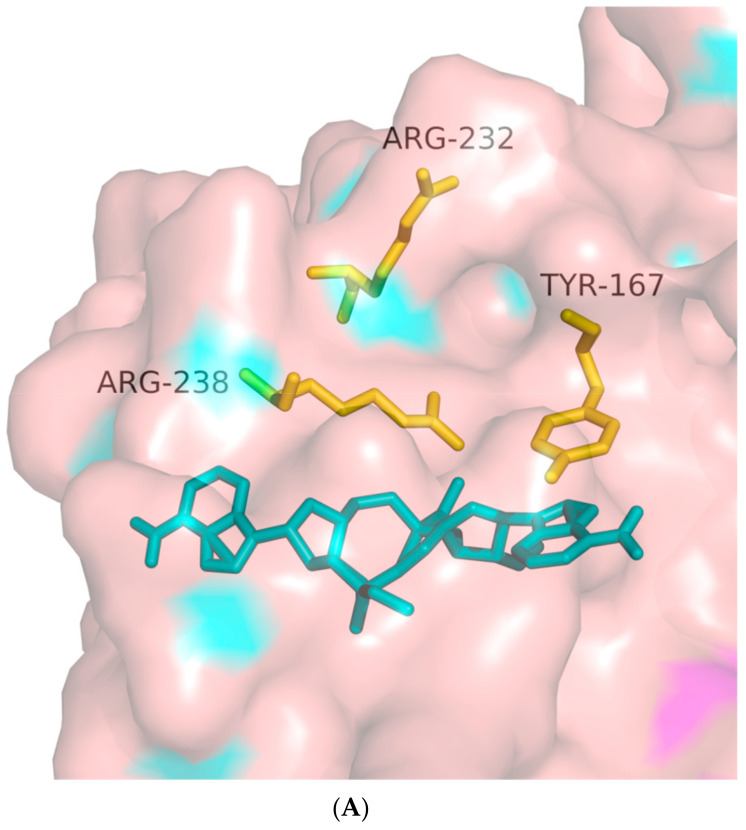
Docking of Thio-Rp, Rp-2′,3-c-di-AMI to the hSTING-WT reactive structural domain. The size of the grid box was set to 10 × 10 × 10. Molecular docking simulations were performed by genetic algorithms with flexible ligands in order to locate the appropriate binding orientation using the AutoDock 4.2 program. (**A**) docking microscopic diagram, (**B**) showing the types of spatial forces between amino acid residues and small molecular functional groups.

**Figure 8 biomolecules-14-00350-f008:**
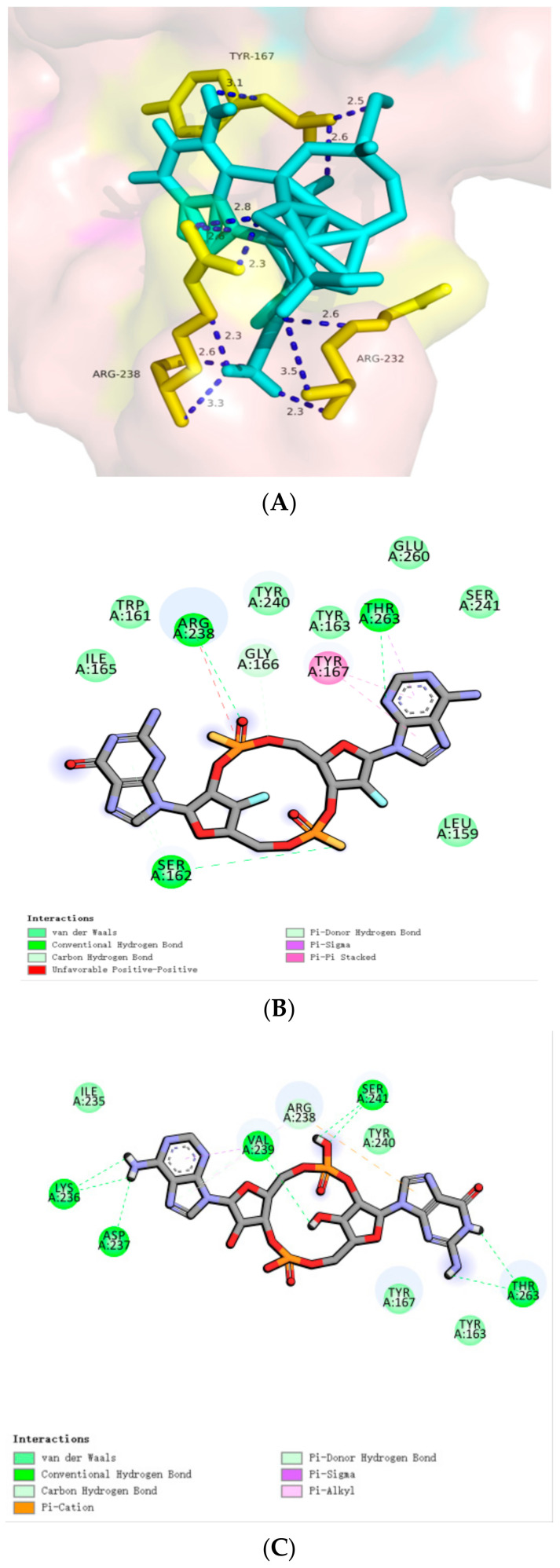
Docking of cyclic dinucleotides to the hSTING-WT reactive structural domain. (**A**) Docking of CDG^SF^ to the hSTING-WT reactive structural domain. (**B**) Docking of MK-1454 to the hSTING-WT reactive structural domain. (**C**) Docking of ribo/xylo-19 to the hSTING-WT reactive structural domain. The size of the grid box was set to 10 × 10 × 10. Molecular docking simulations were performed by genetic algorithms with flexible ligands in order to locate the appropriate binding orientation using the AutoDock 4.2 program.

**Figure 9 biomolecules-14-00350-f009:**
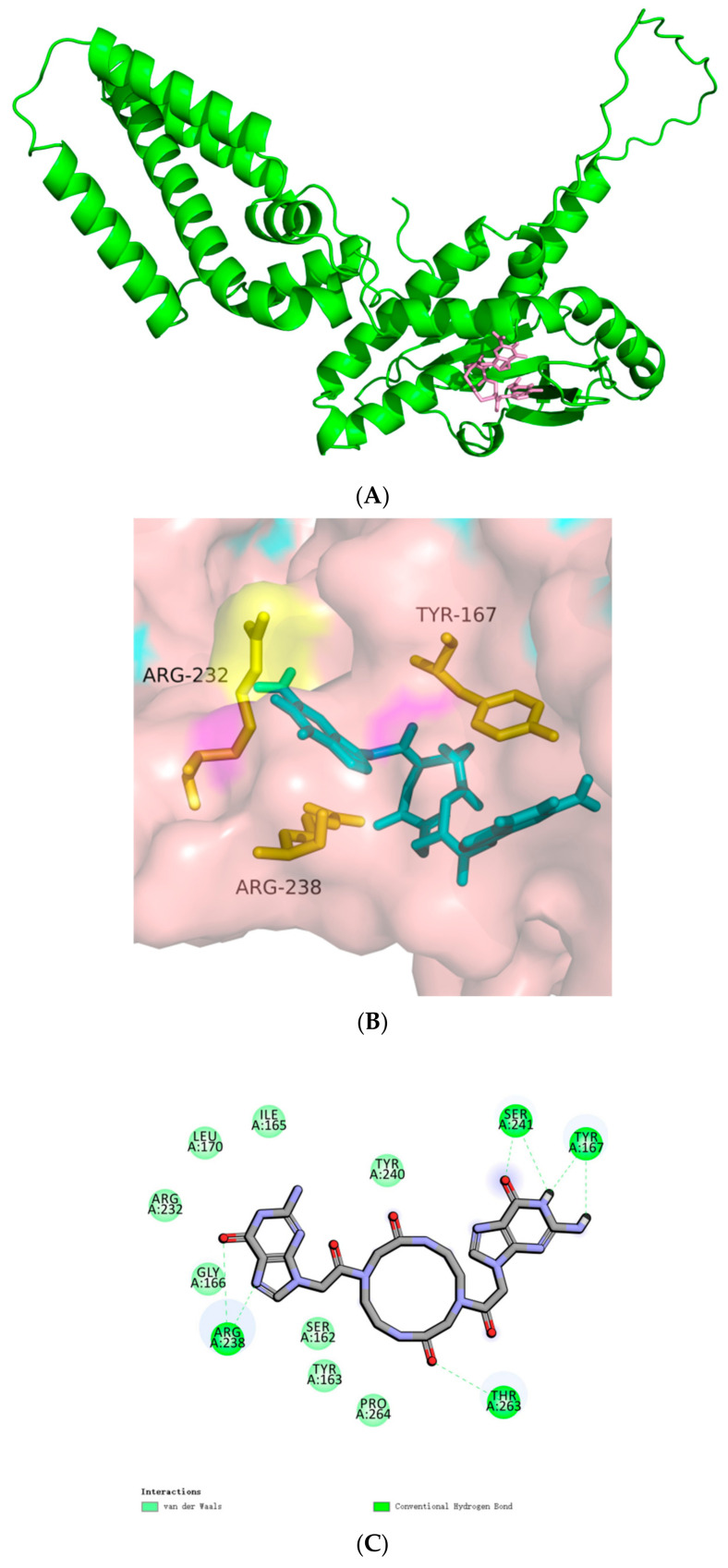
Docking of cyclic dipeptide nucleic acid analogs to the hSTING-WT reactive structural domain. The size of the grid box was set to 10 × 10 × 10. Molecular docking simulations were performed by genetic algorithms with flexible ligands in order to locate the appropriate binding orientation using the AutoDock 4.2 program. (**A**) docking microscopic diagram, (**B**) showing the types of spatial forces between amino acid residues and small molecular functional groups, (**C**) showing the types of spatial forces between amino acid residues and small molecular functional groups.

**Figure 10 biomolecules-14-00350-f010:**
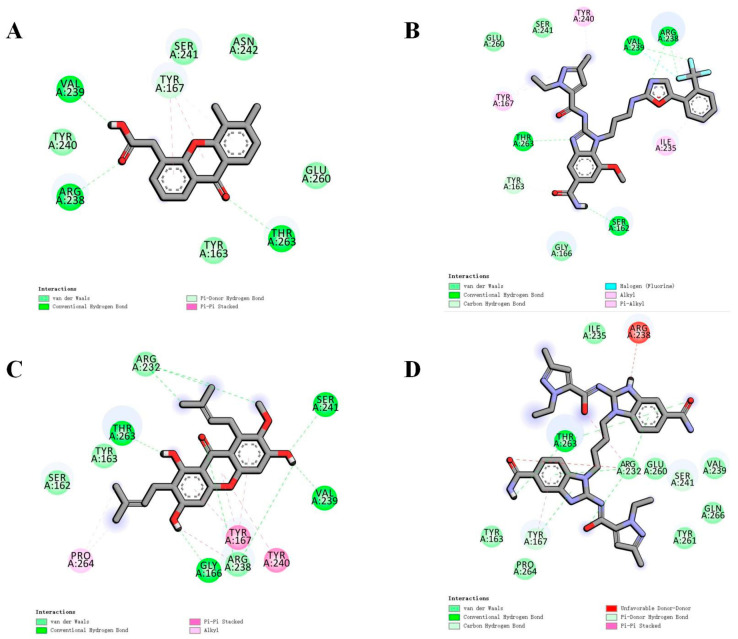
(**A**) Docking of DMXAA to the hSTING-WT reactive structural domain. (**B**) Docking of α-Mangostin to the hSTING-WT reactive structural domain. (**C**) Docking of di-ABZI to the hSTING-WT reactive structural domain. (**D**) Docking of D61 to the hSTING-WT reactive structural domain. The size of the grid box was set to 10 × 10 × 10. Molecular docking simulations were performed by genetic algorithms with flexible ligands in order to locate the appropriate binding orientation using the AutoDock 4.2 program.

**Figure 11 biomolecules-14-00350-f011:**
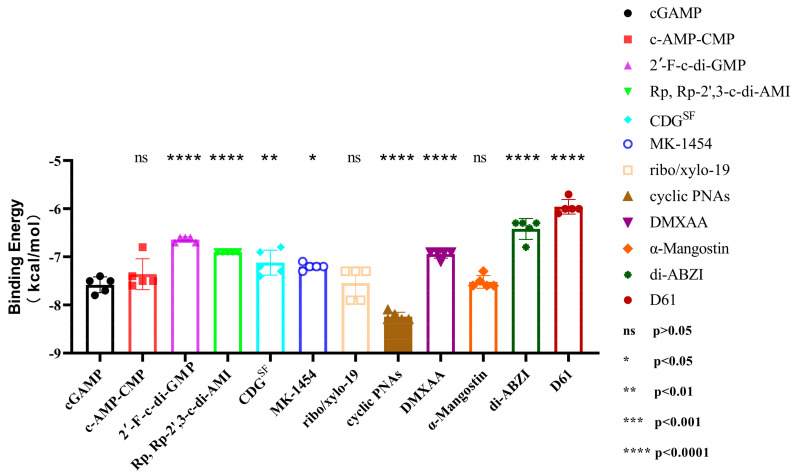
Differential comparison of small-molecule agonists with the classical agonist 2′, 3′-cGAMP. Cyclic PNAs, which we specifically labeled, bind better to hSTING proteins than 2′, 3′-cGAMP, and the results of cyclic PNAs are directly and significantly different from those of 2′, 3′-cGAMP.

**Table 1 biomolecules-14-00350-t001:** Candidate agonist binding affinity to different STING proteins. The red color indicates that the drug binds most tightly to that type of sting protein, which may represent a higher level of biological activity or efficacy.

Enzyme	Candidate Agonists Bind to Different STING Proteins (kcal/mol)				
	WT	R232H	R293Q	AQ	HAQ
cGAMP	−7.5	−7.8	−7.5	−7.7	−7.4
c-AMP-CMP	−7.6	−7.4	−7.5	−7.5	−6.8
2′-F-c-di-GMP	−6.6	−6.7	−6.6	−6.7	−6.6
Rp, Rp-2′,3-c-di-AMI	−6.9	−6.9	−6.9	−6.9	−6.9
CDG^SF^	−6.9	−7.2	−7.4	−7.3	−6.8
MK-1454	−7.2	−7.2	−7.1	−7.3	−7.2
ribo/xylo-19	−7.3	−7.9	−7.3	−7.9	−7.3
cyclic PNAs	−8.1	−8.3	−8.2	−8.3	−8.3
DMXAA	−6.9	−6.9	−6.9	−6.9	−7.1
α-Mangostin	−7.6	−7.6	−7.3	−7.6	−7.5
di-ABZI	−6.3	−6.8	−6.4	−6.3	−6.3
D61	−5.7	−6.0	−6.0	−6.1	−6.0

## Data Availability

All data generated or analyzed during this study were included in this article’s methods section. Other data that support the findings of this study are available from the corresponding author upon reasonable request.

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
