# Peer review of "Exploration of the Binding Mechanism of Cyclic Dinucleotide Analogs to Stimulating Factor Proteins and the Implications for Subsequent Analog Drug Design"

_biomolecules, 2024, doi:10.3390/biom14030350_

Round 1

Reviewer 1 Report (Previous Reviewer 2)

Comments and Suggestions for Authors

The manuscript by Yuan et al. presents a theoretical study of the binding of cyclic dinucleotide analogues to STING proteins.

The answers provided by the authors allayed any previous doubts. It is my opinion that the manuscript can be accepted in its current form.

Author Response

Thank you very much for your professional advice.

Reviewer 2 Report (Previous Reviewer 1)

Comments and Suggestions for Authors

The authors have addressed the various points raised by the reviewers. There are some minor points that need to be addressed before the final publication of the paper.

In section 2.4 (p. 5, line 151), the authors state "...NVT equilibrium and NPT equilibrium..." which is confusing. What type of simulation was employed for the production run, the NPT or the NVT ensemble? The information provided in the methods section is crucial for the readers to understand the scope and significance of the authors' work. Information that is ubiquitous must be avoided and the language used should be precise.

In figure 10 the authors have included the graphical abstract which is confusing. Please edit the text and figures so as to resolve any of these issues.

Comments on the Quality of English Language

The text needs very minor english editing to improve the quality of the presentation.

Author Response

1.Comment: In section 2.4 (p. 5, line 151), the authors state "...NVT equilibrium and NPT equilibrium..." which is confusing. What type of simulation was employed for the production run, the NPT or the NVT ensemble? The information provided in the methods section is crucial for the readers to understand the scope and significance of the authors' work. Information that is ubiquitous must be avoided and the language used should be precise.Response: Thank you very much for your professional advice to help us improve the quality of our manuscript.NVT simulation in molecular dynamics simulation is an important simulation method, which can be used to study the kinetic behaviour, phase transitions and other problems in different systems.NVT indicates that the system is simulated under the conditions of constant volume (V), constant temperature (T), and constant number of particles (N), and the information of thermodynamic properties and kinetic behaviours of the system can be obtained through the selection of suitable NVT simulation methods and parameters.The NPT equilibrium ensures that the number of atoms n, pressure p and temperature t of the system remain constant. The NPT equilibrium performs not only temperature control but also pressure control. Like NVT equilibrium, NPT equilibrium regulates temperature by adjusting the velocity of the atoms, except that the size of the box in NPT equilibrium can be varied. NPT equilibrium regulates pressure by changing the size of the box, e.g., by expanding the size of the box to reduce the pressure when the pressure of the system exceeds a set value.Both have promising applications in physical chemistry and biochemistry, which can help to further deepen the understanding of the behaviour and interaction mechanisms of complex systems. 2.       Comment: In figure 10 the authors have included the graphical abstract which is confusing. Please edit the text and figures so as to resolve any of these issues.Response: Thank you very much for your valuable feedback on this article. The image in this position is the TOC graphic summary chart drawn for this article, which was placed here due to an editorial error, and we have contacted the editor for modification.

This manuscript is a resubmission of an earlier submission. The following is a list of the peer review reports and author responses from that submission.

Round 1

Reviewer 1 Report

Comments and Suggestions for Authors

Major revisions

In methodology section 2.2, what were the parameters employed for the homology modelling? There is no mention of any relevant data, such as template structures or potential restrictions used.

In section 2.3 Molecular docking the authors do not provide any information on the scoring function and the box dimensions employed for the docking calculations.

In section 2.4 regarding the Molecular Dynamics simulations, please provide information regarding force fields employed for the system. Additionally, the authors should provide information on how the systems were set-up and the steps employed prior to the production run.

In results section 3.1, the authors present the analysis of the homology modelling. Reading the paragraph, it is hard to understand what the purpose was of doing homology modelling. As the authors mention, there are already crystal structures of hSTING, therefore the mutations (single point mutations) can be easily modelled on these structures. Homology modelling is usually employed when we do not know the 3D conformation of a particular amino acid sequence.

In molecular dynamics the authors talk about 10 ns MD simulation time while in the RMSD graph the x-axis shows that the simulation time is 100 ns. Please clarify the issue. Additionally, during RMSD analysis there is no mention of what the reference structure employed for the comparison. The comparison would provide more information if the same reference structure (e.g. wt) is employed for the comparison. Also, the RMSF analysis does not show any significant differences between the isoforms. All structures appear to have a similar pattern with slight variations. The MD simulation would provide more information if it was performed on the complexed structures with a respective ligand but I understand that this may not be part of the scope of this article.

In the docking section of the results the authors have used different grid boxes for the different calculations. This choice may impact the results produced by the docking program. Since the binding site of the protein is defined clearly by the authors, the docking simulations should have the same parameters (active site, grid box) for all molecules for the results to be comparable. Otherwise, the results may be skewed.

In the docking section the authors discuss different small molecules and how they bind to STING but they do not provide any information regarding the 3d structures of these molecules. Did the authors draw the substituted compounds in a software? Did they perform any minimisation or geometry optimization on the compounds? Having an acceptable 3D geometry is important for molecular docking calculations. Please provide the relevant information.

 Minor revisions

In Figure 1 legend the authors report parts (A) and (B) but the figure itself contains only the sequence comparison. Please update either the figure or the figure legend accordingly.

Update Table 1 legend since there are no red-coloured values.

In Figure 2 the legend should be updated since the figure contains two parts (A) and (B). The figure legend should correspond to the actual figure presented in the text.

Comments on the Quality of English Language

The text needs some minor editing. In general the language of the text is precise and adequate to convey the authors' intentions.

Reviewer 2 Report

Comments and Suggestions for Authors

The manuscript by Yuan et al. presents a theoretical study of the binding of cyclic dinucleotide analogues to STING proteins. I think there are a few points that the authors should address:

- It's my opinion that 10ns of molecular dynamics simulation is too short to assess the stability of a protein. The simulations should be longer and run in several copies with different starting conditions. In Figure 3A the time is incorrectly reported, there seems to be an extra zero. 

- The values of the interaction energies in Table 1 are very close to each other, so no conclusion can be drawn. It might be more informative if the interaction energies were mean values and the standard deviations were given.

- On line 158 the full list of the PDB structure should be written.

- The full list of interacting amino acids should be written on lines 204 and 501.

- Lines 510 to 522. In this part of the discussion, the authors give two conflicting opinions from one sentence to the next. It would be interesting if the authors could use their results and the literature to develop a pharmacophore that explains the characteristics a molecule must have to bind to the protein.

- In my opinion, the authors are trying to explain the binding mode rather than the binding mechanism. It's my opinion that the manuscript should be revised in this direction. 

Comments on the Quality of English Language

the English quality of the manuscript is good 

Round 2

Reviewer 1 Report

Comments and Suggestions for Authors

There are no further comments

Reviewer 2 Report

Comments and Suggestions for Authors

In my opinion, the data provided by the authors are not sufficient to support the conclusions of the manuscript. 

It is suggested that the simulation time be increased to at least 100ns and calculations be performed from different initial conditions at least twice to enhance the validity of the simulations. It is suggested that the authors present the results of the analyses claiming the achievement of system equilibrium for the entire length of the calculated simulations. They may then choose to use the balanced part of the simulation for further analysis, drawing conclusions and making assumptions. It is worth noting that the current presentation of the results does not allow for the conclusion that the protein is stable at 300K.

Longer simulations could be conducted to further investigate whether the areas of the protein involved in binding small molecules undergo conformational changes or have different conformational freedom than the wild type (WT) protein.

It is important to note that the presented data are insufficient to make any definitive assumptions about the binding mechanism or any other interaction relationship. Presenting the data in Table 1 as an average and standard deviation can help reinforce the conclusions drawn.

There are several methods available for analyzing protein-small molecule complex structures, and it may be worth exploring additional possibilities. 

It appears that there is an error in the reported time in Figure 3A.